# Reuse of Historic Buildings in the Medieval City of Rhodes to Comply with the Needs of Sustainable Urban Development

**Dimitris Giannakopoulos** [1,*], **Zografia Karekou** [1], **Elli Menegaki** [1], **Elisavet Tsilimantou** [1], **Charalabos Ioannidis** [2], **Eleni Maistrou** [3], **Antonios Giannikouris** [4] **and Antonia Moropoulou** [1]

1　School of Chemical Engineering, National Technical University of Athens, 15772 Athens, Greece; zografia.karekou@gmail.com (Z.K.); ellmenegaki@gmail.com (E.M.); eltsilim@mail.ntua.gr (E.T.); amoropul@central.ntua.gr (A.M.)
2　School of Rural, Surveying & Geoinformatics Engineering, National Technical University of Athens, 15773 Athens, Greece; cioannid@survey.ntua.gr
3　School of Architecture, National Technical University of Athens, 10682 Athens, Greece; elmais@central.ntua.gr
4　Technical Chamber of Greece, Amarantou, 28hs Oktovriou, 85100 Rhodes, Greece; antonisgiannikouris@gmail.com
*　Correspondence: dgiannak@mail.ntua.gr

**Abstract:** This paper illustrates a study for the reuse of selected historic buildings in the Medieval City of Rhodes in Greece. This study contributes to the understanding and interpretation of the chronological sequence of major intervention programs for the preservation of the Medieval City. The main idea of the project is the spatial distribution of compatible and various uses to reveal the unique character of the Medieval City. Spatial planning is proposed within the preserved urban zones in order to optimize and harmonize the selected uses according to the needs of sustainable urban development. Criteria to assess the compatible uses that ensure the features and architectural characteristics of the preselected historic buildings, located within the city fabric, were established and validated. The developed methodology that is presented herein and is an indispensable part of a pilot project may be applicable to other cases concerning historic cities. Finally, this paper aims to present a pilot program which promotes the reuse of historic buildings as a part of an integrated preservation plan. Inter-disciplinarity has set the basis for effective policies to guide and control the proposed pilot program, with ultimate objective to ensure sustainable preservation of the Medieval City of Rhodes.

**Keywords:** Medieval City of Rhodes; reuse of historic buildings; GIS processes; sustainable urban development; spatial planning; preservation of historic cities; compatibility; circular economy

## 1. Introduction

The Medieval City of Rhodes [1–3], already declared by the Ministry of Culture as a "Preserved Monument Complex" in 1960 and by UNESCO as a "World Heritage Site" in 1988, constitutes a mosaic of different historic design movements and a site of interest for the scientific field of the protection of monuments.

It is considered a vibrant historic city where urban planning and preservation should be addressed in consideration of the pressure of tourism on the inhabitants and preservation of the city's multilateral character. In terms of reuse strategies in historic cities, there are many theoretical approaches developed by experts in the field of urban preservation, who try to manage touristic activities that have overthrown the traditional urban character of cities, especially in southern Europe [4–9]. The walled city of Mdina in Malta [10], Venice in Italy [11], Syracuse [12] in Sicily or Antalya [13] in Turkey, have been struggling with the terms "museum-cities" or "touristic oriented", creating many problems for the inhabitants as overdeveloped historic cities. In addition, approaches without in-field research such as in Valletta, Malta, have led to the unsuccessful implementation of reuse strategies. A

similar result can be found in cultural heritage cities, where a management plan is entirely absent, for instance in Dubrovnik, Croatia, which suffers from over-tourism. On the other hand, in the northern part, some examples of more successful urban conservation strategies are encountered, e.g., York in England [10] and Bruges in Belgium [11]; however, touristic activities in these areas still represent one of the main sources of economic activity. Finally, regarding the case of the Old City of Corfu, the sole costal city in Greece, where an organized management plan has already been applied, complications are constantly identified that require a redesign of the preservation plan.

## 2. Current Trends—Scientific Background and Approach of the Study

In this study, the process of a reuse strategy is illustrated and analyzed through a specific methodological framework and in line with the current trends, and the results of the field research are documented, classified and utilized in order to compose an adaptive reuse proposal of selected historic buildings in the Medieval City of Rhodes. The developed process of this study (analyzed in the following sections of this paper) consists of three main steps: an adaptive reuse proposal; building evaluation; and a sustainable preservation plan.

### 2.1. Adaptive Reuse Proposal

A theoretical approach of adaptive reuse was only established in the 19th century [2] when Eugene Emmanuel Viollet-le-Duc (1814–1879) recognized adaptive reuse as a way of preserving historic monuments. He argued that "the best way to preserve a building is to find a use for it, and then to satisfy so well the needs dictated by that use that there will never be any further need to make any further changes in the building" [14]. His ideas, however, were strongly objected to by John Ruskin (1819–1900) and his pupil William Morris (1834–1896), who found it "impossible, as impossible as to raise the dead, to restore anything that has ever been great or beautiful in architecture" and instead of restoration, they favored regular care and maintenance to ensure the preservation of historic buildings [15]. In the early 20th century, the conflict between these opposing theories on adaptive reuse was discussed by Alois Riegl (1858–1905) [16]. He ascribed this conflict in theories to the different values their adherences attribute to monuments. Riegl distinguished different types of values which he generally grouped as commemorative values (including age-value, historical value and intentional commemorative value) as opposed to present-day values (including use value, art value and newness value). By including the use value in his assessment of monuments, he recognized the reuse of historic buildings as an intrinsic part of modern conservation [14].

Adaptive reuse, defined by Douglas as "any building work and intervention aimed at changing its capacity, function or performance to adjust, reuse or upgrade a building to suit new conditions or requirements" [17]. The adaptive reuse of heritage buildings is a preservation method used to protect buildings from deterioration and to sustain their value. In addition to extending the lifecycle of the building [18,19], it is also an incentive for creating and sustaining environmental, social and economic values [18], which contributes to a stronger identity [20]. It is considered one of the most important strategies when dealing with heritage buildings and works to achieve a balance between preserving a building and enhancing its role in the urban environment [21]. Research by Bie Plevoets and Koenraad Van Cleempoel [5,22,23] has effectively described different theoretical approaches to the topic of adaptive reuse which have coexisted throughout the scientific debate over the last 50 years: the typological approach, which examines the compatible uses for specific building typologies; the architectural approach; the technical approach; the programmatic approach; and the approach of interior design [12].

The reuse of buildings is one of the most critical factors for the sustainability of preservation operations. Reuse represents the source of post-conservation economies and provides the economic resources needed for maintenance operations that support the sustainability of conservation operations [24]. According to the 'Adaptive re-use of the built heritage, preserving and enhancing the values of our built heritage for future

generations' of the Leeuwarden declaration, adaptive reuse contributes to the building of more resilient and sustainable cities and the application of circular economy principles in the built environment. By re-opening closed or disused spaces to the public, the adaptive reuse of our built heritage can generate new social dynamics in their surrounding areas and thereby contribute to urban regeneration. The reuse of our built heritage can contribute to increasing the attractiveness of areas [25].

The article 'Assessing Cultural Heritage Adaptive Reuse Practices: Multi-Scale Challenges and Solutions in Rijeka' by Nadia Pintossi, Deniz Ikiz Kaya and Ana Pereira Roders [26] demonstrates that the challenges to the adaptive reuse of cultural heritage as defined in the literature are analyzed as follows: availability of reliable information availability of skilled tradesmen and compatible materials [27,28]; compliance with codes and regulation requirements [27,28]; conflict with the local community about the new uses of the heritage [29]; "continuity of local community life" [30]; effective and appropriate community engagement opportunities [30]; economic viability and costs [30–33]; handling of contaminations and hazardous materials [32,34–36]; identification of the new function [37]; minimization of change [30,38]; obtaining the approval of the change of use [27,29,39,40]; "physical restrictions" (e.g., the structural grid) [27,37,38]; political circumstances [41,42]; prevention of values loss [30,31,38]; status of physical decay [43,44]; and finally stigma associated with the building/site/area [45].

As a strategy for heritage conservation, sustainable development and the entry point for circular cities, the adaptive reuse of cultural heritage is receiving increasing attention. However, the implementation of these heritage reuse processes is hampered by several challenges. The overview of these challenges and the proposed solutions raise awareness among the stakeholders involved in implementing heritage reuse, as well as provide evidence to policymakers and decision makers [46].

The reuse proposal in the case of the Medieval City of Rhodes involves the evaluation of selected buildings in a manner that includes: the availability of information; selection of buildings of state ownership; coordination with stakeholders and the community; preservation of the values and economic viability and subsequently the identification of the evaluation criteria for the preferable land uses proposal. These evaluation criteria derive from interconnected stages and are analyzed in Section 4.4. The stages of this research are described in the following chapter and the developed methodology follows the guidelines of the preservation plan process, as explained in the EX.PO.AUS, "Management planning of UNESCO World Heritage Sites—Guidelines for the development, implementation and monitoring of management plans—with the examples of Adriatic WHSs".

### 2.2. Building Evaluation

The historical built heritage represents an important resource consisting of many buildings subject to protection that are in a state of deterioration, neglect or that show significant deficits in environmental performance. Therefore, the architectural heritage, as shown by the Baukultur theories [47,48], requires interventions that look at conservation and simultaneously at performance enhancement and rehabilitation strategies renovation, that contribute to building a sustainable society. Douglas (1996) defined buildings as being heterogeneous, as they are different in their own way including in terms of the location, subsoil conditions and different access provisions. Each building has its own characteristic according to the differences in the internal and the external environments of the building [49].

The evaluation of historic buildings in the context of the present case study begins with the on-site research of the building in the Medieval City of Rhodes and is completed by the evaluation criteria as an outcome of this study. The field research process involves the complete recording of the evaluation sheets, the demarcation of the neighborhoods in maps and the recording of existed land by the research team. The research team completed all evaluation sheets for every historic building, including the historical and architectural

elements, the morphology, the current uses of the building, the alterations, the maintenance status and the type of protection (in cases of the listed monuments).

The on-site research material consists of evaluation sheets, digital maps, photos of the buildings, public spaces and the surrounding area. More specifically, the research team developed maps with by using geographical information systems (GIS), establishing principles, understanding the problems that lead to the current abandonment of selected historic buildings, acquiring information through bibliographical research and creating digital maps. All this information presented through digital maps along with the evaluation sheets were utilized for the development of the evaluation criteria for the reuse proposal.

*2.3. Historic City and Sustainable Preservation Plan*

The term 'sustainable management' of cultural heritage has been included in the Operational Guidelines for the Implementation of the World Heritage Convention since 2005 as a guide for the management of World Heritage Sites. This emphasizes the constructive role of cultural heritage in promoting human development, which in the long run will lead to an improvement in the sustainability of world heritage itself [50].

> *Cities are amalgams of buildings and people. In the urban artifact and its mutation are condensed continuities of time and place. The city is the ultimate memorial of our struggles and glories: it is where the pride of the past is set on display* [51], p. 16

This phrase summarizes the inestimable preciousness of historic cities for humanity. Historic cities are excellent sites of historic memory, as they preserve the complexity of the social, cultural, economic and artistic aspects of past civilizations. Keeping historic centers alive, in the sense of a habitable housing development and not just a passive monumental site, consists of a perception where in the development of sustainable cities active participation of the inhabitants is a prerequisite. In such a model of a city, the cultural heritage of historic centers can be considered as a non-renewable economic resource that needs an optimum exploitation plan.

Historic urban conservation should be based on a governance model which will develop strategies and promote synergies between stakeholders, institutions and residents so that, through participatory processes, cultural heritage can become a common consciousness. Conservation plans must identify and protect the elements that contribute to the values of the historic towns, as well as the components that enrich and demonstrate their uniqueness [12].

Decisions made without a preservation plan can be dangerous because of the complexity of existent heritage (Figure 1). Decisions regarding heritage management that do not take into account a preservation plan are considered a dangerous practice, since such use cases usually involve complex heritage sites. Serious conflict can arise from a lack of apprehending the values of a site or of the management dynamics of cultural heritage works. Other issues arise from the exclusion of a key discipline expert from the planning process. "There will be no urban future—less so sustainable urban development—without a full understanding of the power of culture in addressing the social needs of city dwellers and their aspirations to a better quality of life [52], p. 17".

It is self-evident that the spatial form of a historic area dictates planning and adherence to the existing traditional morphology, thus avoiding drastic changes and interruptions in the continuity of the urban fabric. It is also to be expected that the traditional uses continue to function as they constitute elements of a historic area's identity [53].

According to the UNESCO World Heritage Committee, as referenced by the Comprehensive Preservation and Management Plan for the World Cultural Heritage by the Okinawa Prefectural Board of Education, the preservation plan is defined as follows: a plan that clarifies the relations among component parts (individual cultural properties) and sets out policies and measures for comprehensive preservation and management in an overarching context. In order to ensure the appropriate protection of such a nominated property consisting of diverse component parts, a comprehensive preservation and management plan is necessary, which not only addresses the preservation and management of individual

component parts, but also sets out the policies and measures for the integrated protection of the whole nominated property, and takes into account the identified interrelations among the component parts [54].

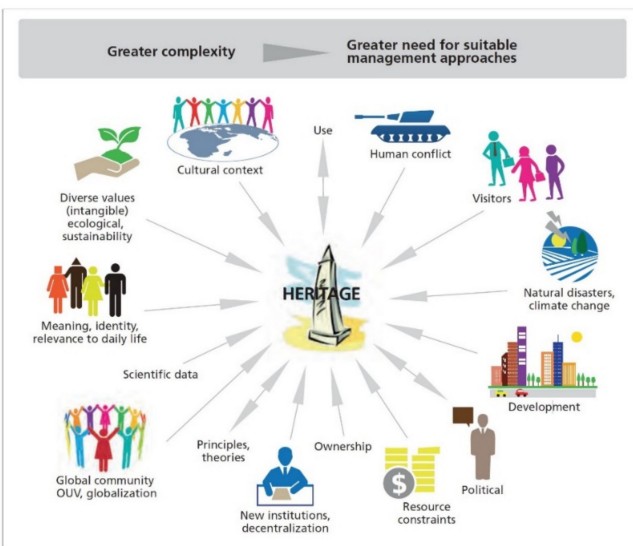

**Figure 1.** Examples of old and new issues in heritage management [Adapted from World Heritage Resource Manual, UNESCO, Ref. [7]].

The development of a management plan may have the following stages: Stage one—preparation; Stage two—data/information gathering; Stage three—significance/condition assessment; and Stage four—developing responses/proposals. It is also considered a tool for establishing a dialogue with the stakeholders and also for protecting the values of a heritage site [55].

## 3. Materials and Methods

### 3.1. The Case of the Medieval City of Rhodes

In the case of the Medieval City of Rhodes, a pilot program for the reuse of historical buildings was formulated which concerns the renovation of buildings with the reciprocal performance of new uses. The program will contribute to the further development of circular economy initiatives [56]. Within this framework, a study for the reuse of selected historic buildings in the Medieval City of Rhodes is presented in this paper. The aim of this study was to contribute to the understanding and interpretation of the chronological sequence of major intervention programs for the preservation of the Medieval City. In addition, this paper presents the evolution from the on-site investigation of the historic city to the current need to draft a preservation plan.

More specifically, the on-site research material constitutes the input for the building evaluation process. All selected buildings are documented into evaluation sheets, as well as their surrounding area. The main scope of the field research is the listing of the evaluation criteria, which acts as the main tool of the reuse proposal. The next step is the development of evaluation criteria, which will serve as the guidelines to the proposal of land uses, which is the final outcome of this research and the proposal of the reuse strategy.

The present study highlights the need to set up an integrated conservation plan and proposes a reuse strategy as an important tool for the revitalization of historic cities. The architectural survey begins with the preparation (establishing the principles) and the record of the situation (documentation/recording of the urban/residential zone-data gathering), the identification of the monumental wealth, the validation of the damage (current state of deterioration) and the need for intervention (condition assessment–conservation/intervention processes). The next stages are the preliminary and final proposals (architectural, structural study) for the reuse of selected buildings that are a property of the Greek state (public

property/government-owned property). The last step includes the implementation (execution of the project) of the proposed intervention processes and the continuous monitoring of the structures and infrastructure.

The reuse strategy proposes a more comprehensive redevelopment of the area and a functional modernization of the infrastructure. The Medieval City of Rhodes needs to stay connected with the modern ways of living, while remaining a unique vibrant city primarily for its residents but also for the tourist community.

*3.2. Main Concepts—Conceptual Framework of the Reuse Proposal*

This paper presents a study within a methodological framework in terms of a reuse proposal, including the use of geographical information systems. The evaluation of selected buildings in the historic city of Rhodes is proposed and an adaptive reuse proposal of land-uses within the certain context of a general sustainable preservation plan is presented. The conceptual framework of the reuse proposal can be described in the following scheme (Scheme 1):

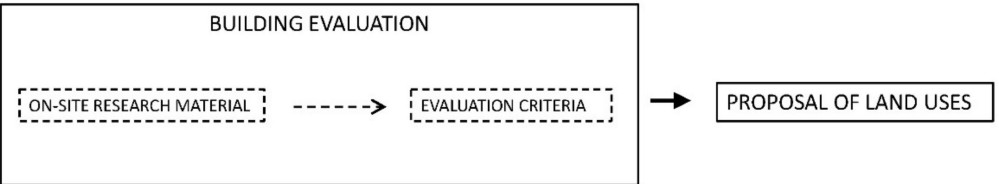

**Scheme 1.** Conceptual framework of the reuse proposal.

The framework of the proposed methodology sets the on-site research material as an input to the building evaluation process in order to output the evaluation criteria, which defines the proposed land uses of every historic building. The final outcome of the whole process is the reuse proposal map of land uses.

In order to articulate an adaptive reuse proposal of selected buildings that belongs to the Greek state, a specific process is required. In this term, a research question is formulated as follows: How does an adaptive reuse proposal of land uses in the Medieval City of Rhodes, as an outcome of selected buildings evaluation, contribute to a sustainable preservation plan of the historic city?

3.2.1. Reuse Proposal

According to Prof. A. Moropoulou [56], it is estimated that there are currently over 300 historic buildings in the Medieval City of Rhodes which can be restored and used (source Ephorate of Antiquities of Dodecanese). Some buildings are selected under the reuse program (Figure 2), in order to propose a complete scenario of adaptive reuse in the historic Medieval City of Rhodes. These buildings are owned by the Greek state [57] (Hellenic Corporation of Local Development, Ministry of Finance, Ministry of Culture, Archaeological Resources Fund). Their restoration and utilization will be planned and implemented within the framework of a program contract between the Greek state and the Municipality of Rhodes, in order to serve the purposes of the program.

The reuse proposal aims to meet the needs of the protection and promotion of monuments and the creation of a mechanism to control the state of preservation and restoration throughout the Medieval City of Rhodes. It introduces a preliminary adaptive reuse for 86 selected buildings in the Medieval City of Rhodes (Figure 3).

The functional reintegration of buildings into social and productive activity is proposed to be achieved with uses that fall into the four following categories:

(A)  Tourist accommodation and leisure activities;
(B)  Commercial areas;
(C)  Workshops of traditional arts and culture;
(D)  Social housing and services.

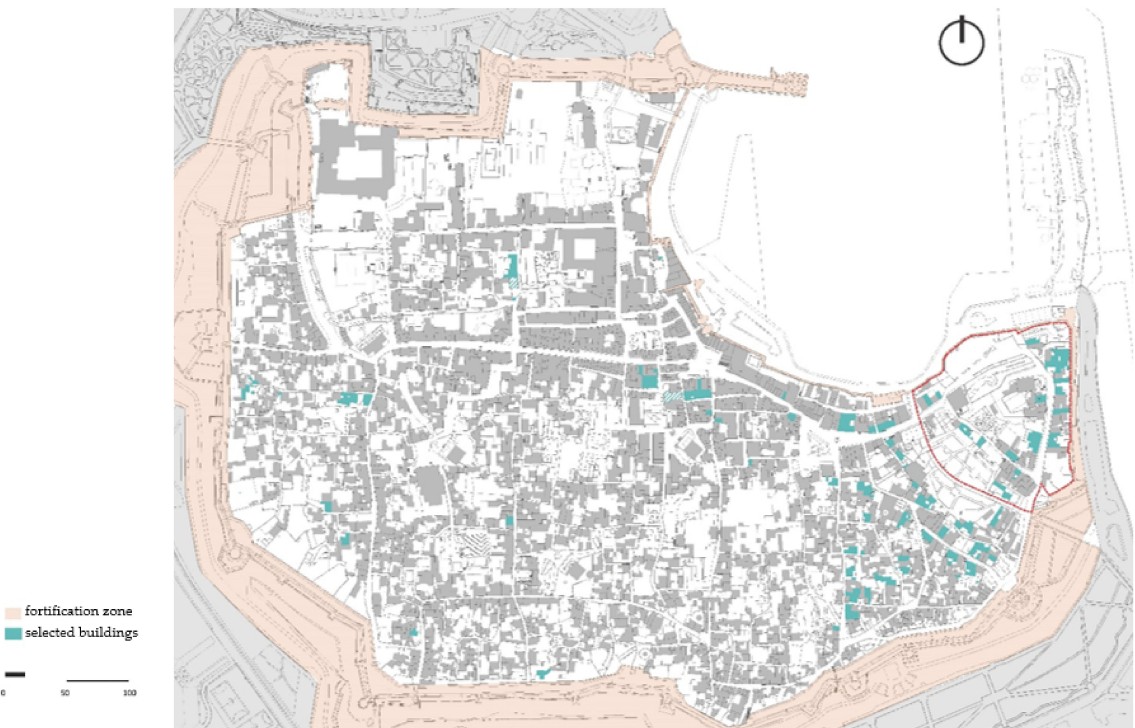

**Figure 2.** Selected buildings of the reuse proposal.

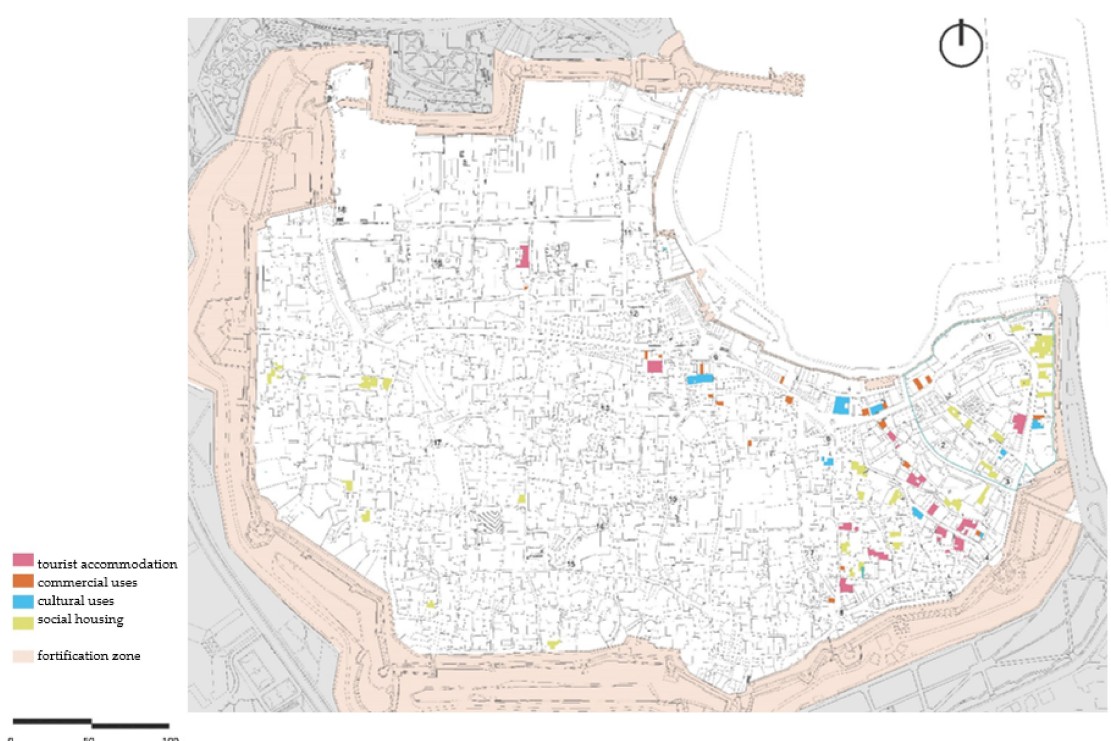

**Figure 3.** Proposed land uses of the selected buildings.

### 3.2.2. Sustainability of the Reuse Proposal

The proposal is presented as applicable and can be characterized as a pilot project in accordance with the principles of integrated protection. It should not be a museum that only functions for tourism [58]. Awareness of the importance of the physical context, built and natural, in the urban conservation and management process has enabled a redefinition of the relationship between the component parts of the city and its region, a necessary step

to ensure the quality of urban spaces and respect for social needs. It is essential to focus on the particularities of historic centers and analyze the changes that led to these areas' status today [59].

Historic conservation should be integrated in national development policies and agendas and involve local authorities and communities, where public service providers and the private sector cooperate. Historic urban conservation should be based on a governance model which will develop policy development strategies and promote synergies between educational institutions and residents so that, through participatory processes, the identity of the place can be recognized, and the cultural heritage can become a common consciousness [24]. The reformation and exploitation of historic buildings, monuments and archaeological sites necessitate a process ensuring conservation studies and restoration works following a holistic approach. Consequently, integrating historic sites into a city context with a pedestrian zones network, as well as their reformation/renewal through urban and architectural interventions, develops a coherent whole which reveals the city from an esthetic point of view. Historic conservation should be the subject of a holistic approach that should include many different aspects [60].

The key elements of urban planning are usually clustered in street grids, public areas, building styles, land uses and other factors. The initiatives of planning in a space with cultural heritage values, however, should be oriented to include cultural heritage conservation as a vector of sustainable development [32]. Indeed, it is necessary to understand sustainability in a holistic sense to ensure sustainable development while respecting the characteristics of a historical environment, requiring a correct understanding of elements, approaches, methods, tools and techniques for the evaluation of a project environmental impact.

This study aimed to adapt the reuse proposal to the existing environment by taking into account all public spaces and park zones (Figure 4). Through the on-site research, all green spaces were recorded. The proposed uses are placed so that the few green spaces are utilized by all users, adding a local special character to each park. For example, the use of social housing around a park, promotes the local character of the neighborhood without removing it from public use. Cultural heritage management is directly related to the economic viability of monuments. Today, the aim of sustainable cultural heritage versus the cultural heritage museum requires the activation of the public sector in order to highlight the direct and indirect positive effects of conservation [38]. The adaptive reuse of cultural heritage could provide environmental, economic and social benefits to local communities [61].

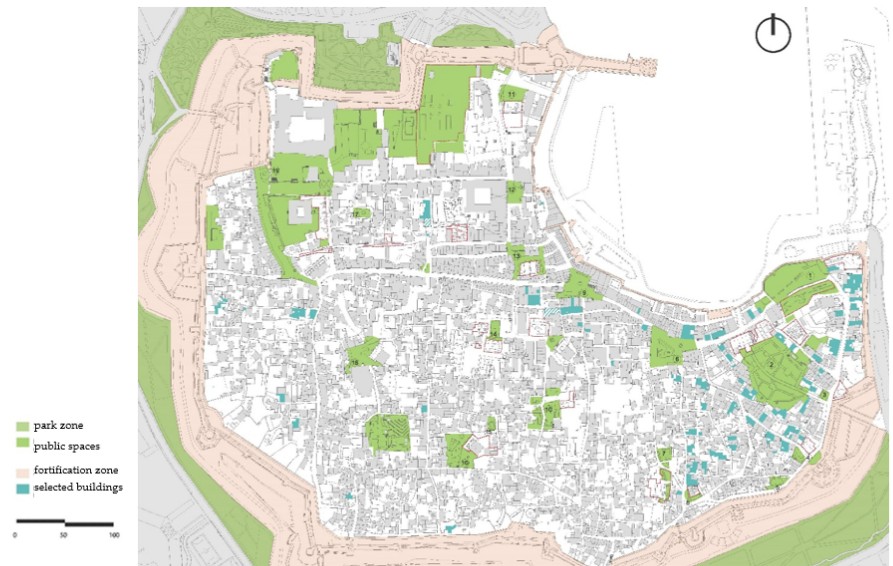

**Figure 4.** Public spaces and parks of the Medieval City.

*3.3. Study Area Analysis*

3.3.1. Medieval City of Rhodes—Historical Information

The architectural styles of the Medieval City of Rhodes that are encountered today contain characteristics from the Byzantine, knighthood and Ottoman traditions and can be classified into the following categories: (a) architectural style of the knighthood period; (b) style of the Ottoman period; (c) neoclassical of the Ottoman period; and (d) the mixed style (medieval buildings with additions). Therefore, the knighthood, Ottoman, neoclassical and popular buildings coexist.

The Medieval City of Rhodes is considered a formal example of a diachronic Mediterranean city adapted in the European model, which characterized the formation of the medieval urban historical centers. The harbor of Rhodes represented an important turning for the eastern Mediterranean, as it was favorable for the grounding of boats with tradesmen and pilgrims travelling to and from the Holy Land (Figure 5a). The conquering of the city of Rhodes by the Ioannite Knights of the Order of St John of Jerusalem (or "Knight Hospitallers") in 1309 signaled a new era. The transformation of the urban built environment (Figure 5b), is characterized by the sovereign elements of fortified west European medieval cities, but has to date maintained the memories of the big Hellenistic Hippodamian plan and of the Byzantine city which it succeeded. The Medieval City of Rhodes is characterized as a center of economic and cultural influence of our era at the crossroads between east and west [62].

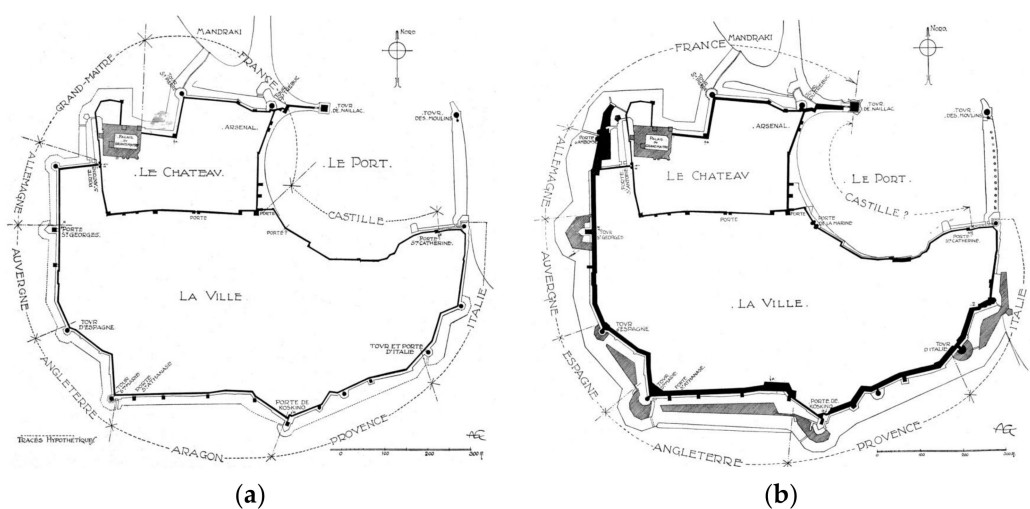

(**a**)                                                                 (**b**)

**Figure 5.** Explanatory map of the medieval City: (**a**) structures and combat posts in 1465 and (**b**) in 1522 Adapted from Ref. [62] 1921, Gabriel A.

The protection of the medieval city dates back to the Italian occupation [1] when in the masterplan of the city of Rhodes (1926), its walled part was defined as a "Monumental Zone" which included extensive parts of land around the fortification as a control area. The Dodecanese Military Command, appointed after the liberation in 1947, immediately took care to protect the medieval city from uncontrolled demolition. In 1948, 239 medieval, Muslim and newer architectural buildings inside and outside the walls were declared "historic monuments". In the 35 years following the integration of the Dodecanese until the mid-1980s, the population had been steadily declining and the gradual decline of the Medieval City within the new city of Rhodes—which showed steady growth—was evident. Ruins from the bombings of 1945 and from the strong earthquake of 1957 along with the lack of infrastructure degraded the Medieval City into an area of cheap housing, and at the same time, created a wave of residents fleeing from it.

From the Italian occupation and the rich history of the decades that followed, a program contract YPPO-TAPA- (Ministry of Culture—Archaeological Resources & Expropriations Fund) Municipality of Rhodes (1985–2005) was developed. In 1982, the Deputy

Ministry of Youth of the Ministry of Culture submitted a proposal to UNESCO. The Ephorate of Antiquities of Dodecanese prepared and submitted the necessary documents in order to include the Medieval City of Rhodes in the World Heritage List. Following an ICOMOS evaluation [63], UNESCOs' World Heritage Committee enlisted the Medieval City of Rhodes in the World Heritage List in 1988 [64].

The objective of the twenty-year Program Contract (1985–2005), was to take action for the preservation, protection and promotion of the monuments of the Medieval City and to take rescue measures in dilapidated monuments, buildings and archaeological sites. The work performed during the twenty years of the Program Contract was extremely productive. Today, the medieval city is protected by a modern institutional and legislative framework, including the selection of the city in the UNESCO list.

### 3.3.2. Present-Day Built Environment

The complexity of the issues in the process of keeping a historic city alive, where there is a need to modernize the infrastructure in parallel with solving the problems created by tourism development, has increased the need for an integrated conservation plan. It is self-evident that the spatial form of a historic area dictates the planning and adherence to the existing traditional morphology, thus avoiding drastic changes and interruptions in the continuity of the urban fabric [65]. It is also expected that the traditional uses continue functioning as they constitute the elements of a historic area's identity [66].

The Medieval City of Rhodes is mostly built with local porphyry (brownish-yellow sandstone limestone), which has been traditionally built on carved or semi-hewn stone structures [1,67]. The buildings are framed with pebble floor coverings in the external common areas (courtyards) and in several parts of the roads. Wood was an important part of the construction, as an element of the load-bearing structure. During the Ottoman period it was a structural element in the facades (loggias, sahnisia). The field research about the present-day built environment of the Medieval City of Rhodes is divided into four different orientation uses:

(a)   The demarcation of neighborhoods and priority zones and the documentation of areas of interest (Figure 6);

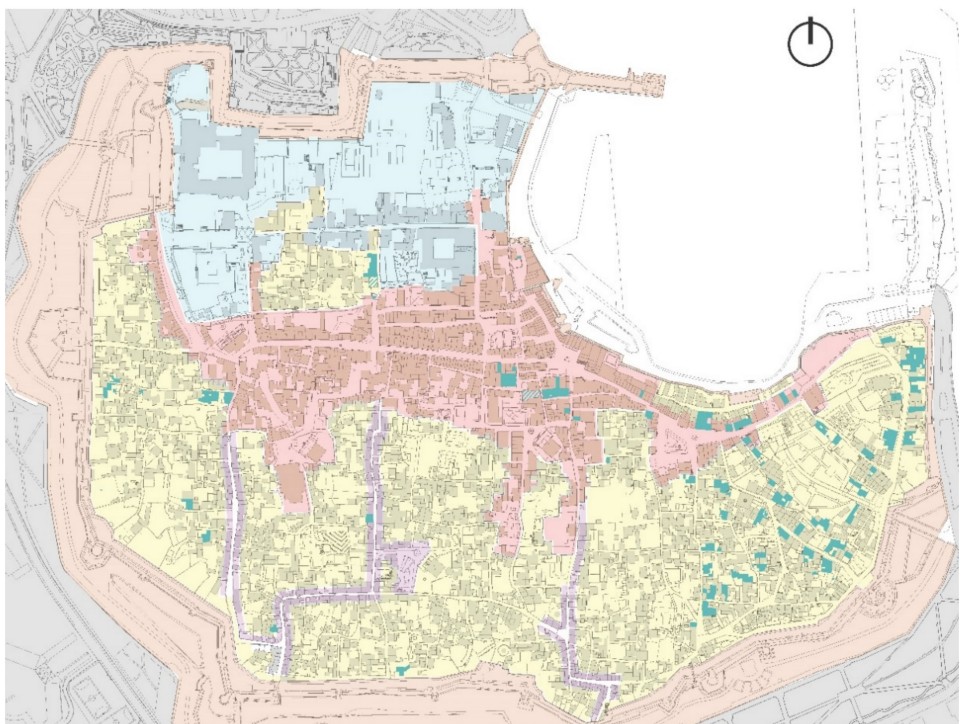

**Figure 6.** Zones of existing the land uses of the Medieval City.

(b) The marking of existing land uses (ground floor—(Figure 7); 1st–2nd floor—(Figure 8)) and the marking of the problems of the existing situation;

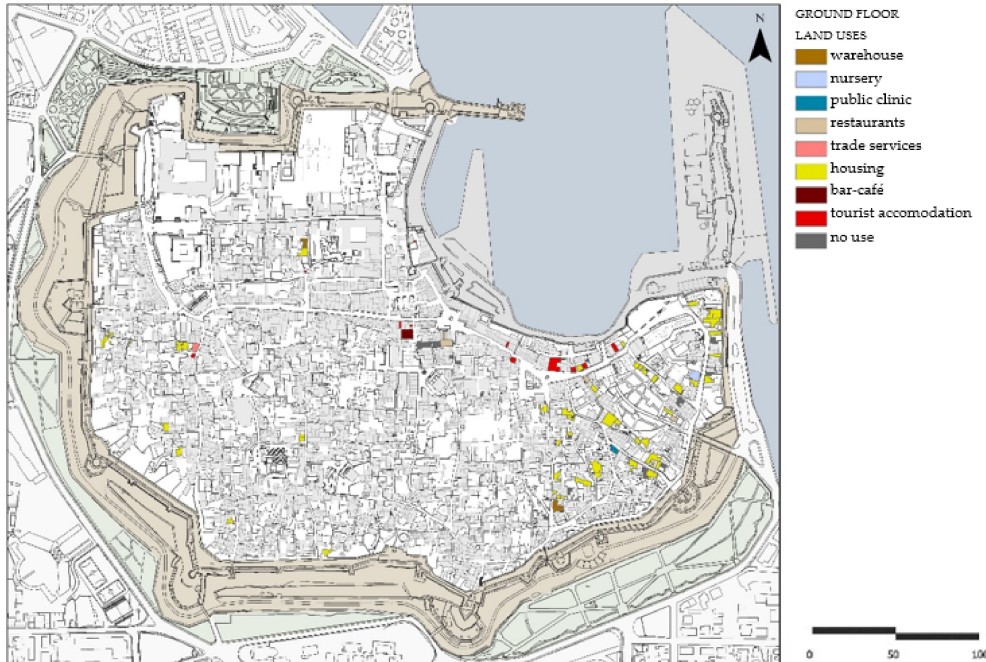

**Figure 7.** Land uses (ground floor).

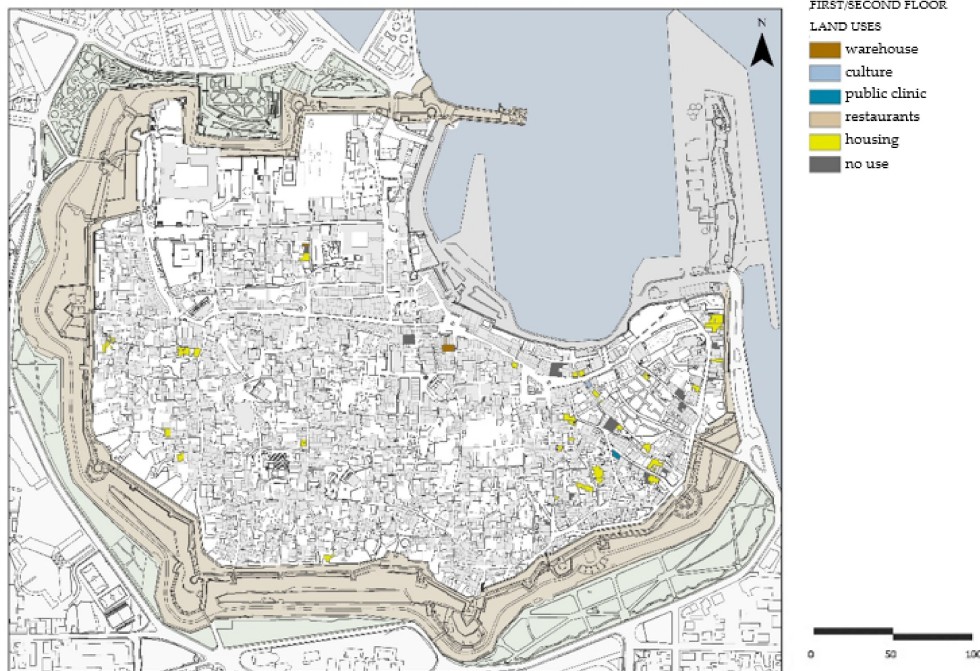

**Figure 8.** Land uses (1st/2nd floor).

(c) The recording of the existing restored buildings (Figure 9), interventions and renovations and the highlighting of the problems that appear;

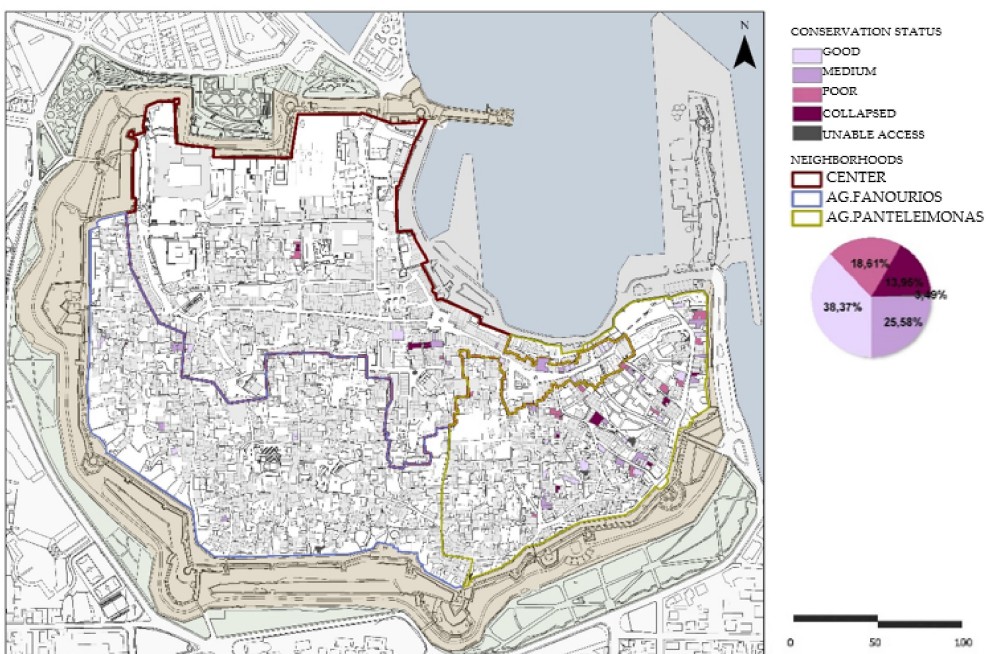

**Figure 9.** Existing maintenance status (good–medium–poor–collapsed).

(d)    The registration and identification of the buildings that need protection and/or restoration intervention based on the following evaluation criteria (as analyzed in chapter 4.4): (a) the property status; (b) the location; (c) the existing use; (d) the coverage; (e) the typology; (f) the conservation status; (g) the morphology assessment; (h) the proposed use; and (i) the proposed degree of intervention (intervention category).

## 4. Results—Phases of the Reuse Proposal

The reuse proposal analyzed in this paper includes five (5) different phases (Figure 10). The architectural survey constitutes the first phase, namely the preparation (establishing the principles). The second phase is the on-site research which includes the record of the situation in evaluation sheets (documentation/recording of the urban/residential zone-data gathering). The third phase includes the GIS creation. In this phase, all the material from the on-site research is collected and imported to a database. In particular, the aim of the GIS creation is to organize and clarify (i) the identification of the monumental wealth; (ii) the validation of the damage (current state of deterioration); and (iii) the need for intervention (condition assessment–conservation/intervention processes). One of the most important phases of the proposal is the 4th phase that consists of the setting of the evaluation criteria, which leads to the final outcome of the reuse land proposal map. This phase constitutes the final proposal for the reuse of selected buildings belonging to the Greek state (public property/government-owned property). The last phase (fifth phase) is the adjustment and execution of the proposal and the continuous monitoring of the structures and infrastructure.

### 4.1. Preparation—Establishing the Principles (Phase 1)

The principles for the conservation should be based on a process, which will highlight the value of the inherited urban fabric as a component of urban sustainable development [68]. It is also necessary to keep the existing traditional uses as they define the way of life of local communities. Guiding land use principles constitute the backbone of the holistic approach for the preservation plan. They serve as a general framework which will guide future land use decisions. Compatible land uses in urban areas are planned by providing a type and mix of functionally well-integrated land uses which meet the general social and economic needs. Conservation plans must identify and protect the elements contributing to the

values and character of the town, as well as the components that enrich and demonstrate the character of the historic town and urban area [69].

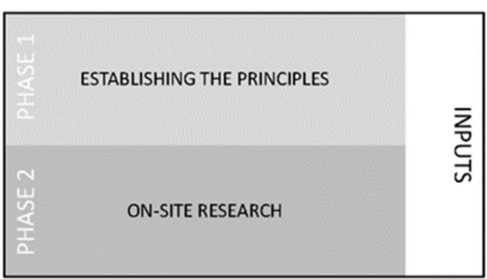

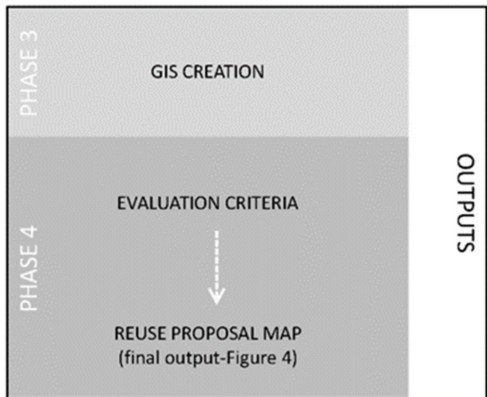

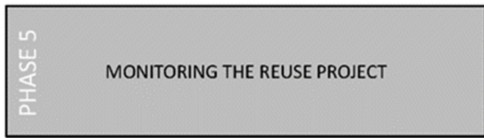

**Figure 10.** Diagram illustrating the phases of the reuse proposal.

### 4.2. On-Site Research Material—Evaluation Sheets (Phase 2)

Data from the survey (including both on-site recording and research) were gathered into evaluation sheets. In order to evaluate a building for the reuse proposal, it is necessary to have a familiarity with the architectural morphologies, the date of construction, the location and the construction elements. The form of the evaluation sheet (Figures 11 and 12) was prepared according to the evaluation criteria that were established. The main categories that constitute the sheet are:

- Main information (street, house/flat number, city block, quarter of Medieval City, registry piece);
- Uses (initial–present);
- State of use (abandoned–rent–squat);
- Short description (number of floors, coverage, building area, roof type, yard and basement).

As illustrated in Figure 11, there are four categories in a grade system (good, medium, poor, collapsed) as established by the researcher's team. The use of grades allows objectivity in terms of assessment. These may sometimes deceive by appearing even more precise than they actually are: nevertheless, they represent the most useful available measuring tool if set out rationally and sensibly. Verbal grades (excellent, very good, etc.) are the most suitable. The evaluation sheet should be drawn up in a manner that enables graders to circle the appropriate grade opposite each criterion. Space should be left so that concise reasons for the grades may be written to the side. The evaluation sheet should be attached to the form that has been used to survey the same building [66]. Additional analyzed information completes the evaluation sheet, including a description of the alterations transforming the original morphology, a brief mention of the alterations that are installed

and some comments. Finally, analyzed morphological documentation is described as the final description analysis of the selected building (Figures 11 and 12). Systematic knowledge of the buildings being studied is needed before rational evaluations of the buildings can be made and before decisions about their possible conservation can be reached. For the detailed survey, as much information as possible about the history, status, condition, character and context of every building should be compiled [66].

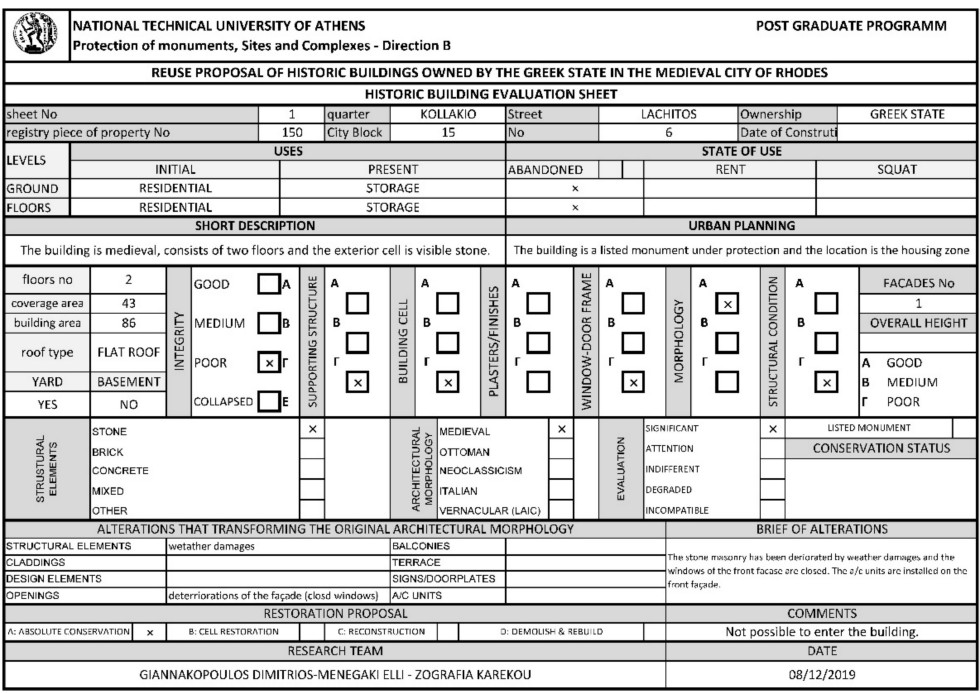

**Figure 11.** Example of an evaluation sheet of selected buildings: front page—short description.

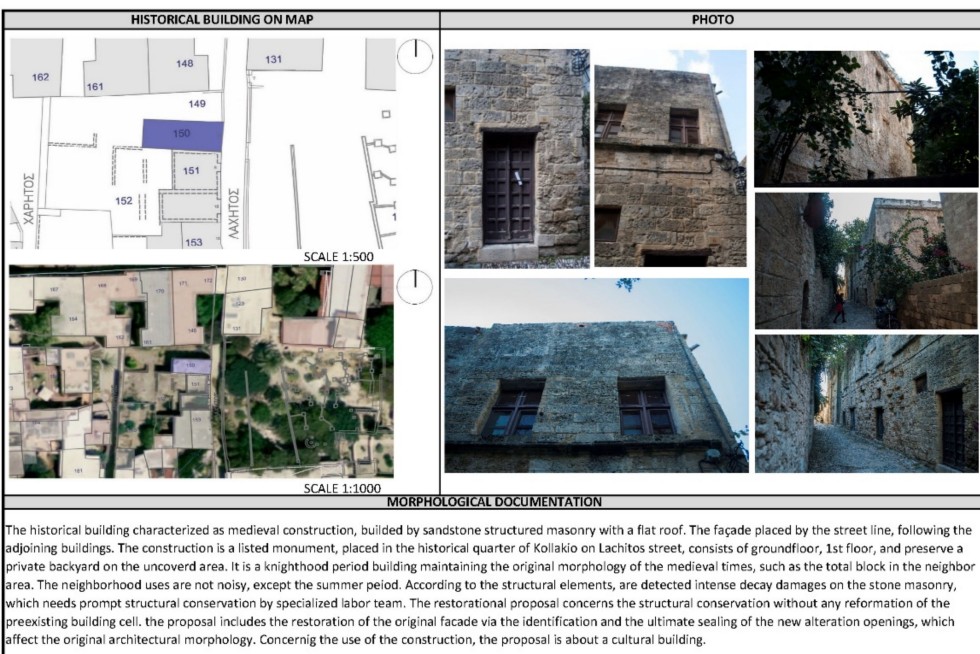

**Figure 12.** Example of an evaluation sheet of selected buildings: second page—description analysis.

*4.3. GIS Creation (Phase 3)*

The recording and geographical representation of the historic buildings (owned by the Greek state) of the Medieval City of Rhodes was carried out in the software program

ArcGIS 10.3.1. by ESRI, while at the same time, AutoCAD Map 3D 2012 of Autodesk was used. The plan of the Medieval City was used as a linear background, with a scale of 1: 1000, georeferenced with the Hellenic Geodetic Reference System 87. While importing the recorded data into the building tabs, a database was created which was then registered in the ArcGIS program and connected to the entity of the buildings. Thus, the GIS system included data on the eighty-six buildings collected from the on-site survey. A database was created and connected to the acquired spatial information in ArcGIS 10.3.1. Various maps were created and useful information was drawn regarding the current state of preservation of the buildings and criteria were set for their reuse while maintaining and preserving their values.

The fields were filled with information regarding the building's geolocation (cadastral portion number, district, street, number, OT), the use, the housing status, the number of floors use (ground floor and first floor), the coverage area, the type of housing, the materials of the supporting organization, the existing conservation status, the morphological alterations, the rhythmology, an overall evaluation, proposed additional interventions and the declaration of certain buildings as monuments. The use of GIS in the present study enabled the storage of large volumes of collected data and linked them to geographical entities. The database created contains additional fields that were not completed in the present study, enabling future processing and use. At the same time, the way it was structured enables an ordinary user to search for information about the building of interest.

*4.4. Evaluation Criteria (Phase 4)*

The evaluation criteria that compose the proposal of the land uses in the historic town of Rhodes navigate the compatibility and the appropriateness of reuse. The well-tempered distribution of the reuse type depends on the documentation of the survey according to the on-site observation and to every evaluation sheet of each selected building (Figures 11 and 12). The evaluation criteria can be classified into nine (9) categories:

(a)    The Property Status

All listed buildings belong to the Greek state. However, the different states of use for each separate building creates a complexity at the reuse proposal (Figure 13). There are many abandoned buildings, illegal squats and houses rented by locals.

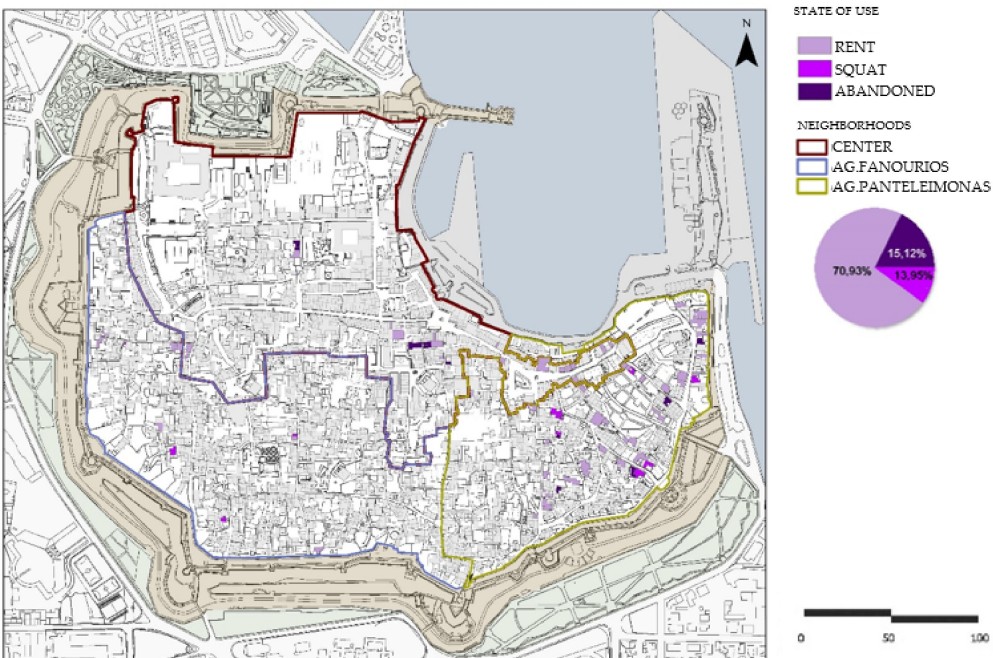

**Figure 13.** States of use (rent, abandoned, squat).

Listed buildings are complex as they have a special statute and the intention of the developed program is that they will be leased for a period of 25 years to a party which will use it privately, in a way that is compatible with the continuous build-up of the Medieval City of Rhodes as a blend of the old with the leading edge of today. Examples include international organizations, diplomatic offices and the consulates of foreign countries or private enterprises active in technology, media, advertising. It is estimated that the procedures for the award of the "design, build and operate" leasing contract, the architectural and engineering designs and the renovation itself will require a period of 18 months.

The current state of use affects the pathology of the listed buildings. In the case of illegal squats, buildings are mostly ruined and abandoned. Rented houses, in contrast, are well preserved by the residents and also by the local community. Generally, it can be observed in documentation that the state of current use (Figure 13) affects the condition of buildings and hence the proposal of their reuse.

Regarding the property status of the selected buildings, this selection was made since the rehabilitation project of the Medieval City of Rhodes includes the Municipality of Rhodes. The program agreement for this future development is between the Ministry of Culture and the Municipality of Rhodes. The third parties that will participate and will be engaged in the program design and implementation are: (a) the Fund of Archaeological Proceeds, TAP; (b) Local Development Enterprise of Rhodes—DERMAE; and (c) National Technical University of Athens—NTUA. The responsibilities of each partner will be clarified in the context of the framework agreement between the Ministry of Culture and the Municipality of Rhodes.

(b)    The Location

The character of the surrounding area of the building is crucial for the reuse of buildings and depends on each neighborhood (Figure 14). Existing land uses zones in the Medieval City of Rhodes that characterize the location of every building are housing, central-commercial uses, cultural-public services and local neighborhood. Generally, the land uses refer to three correlative zones (Figure 6).

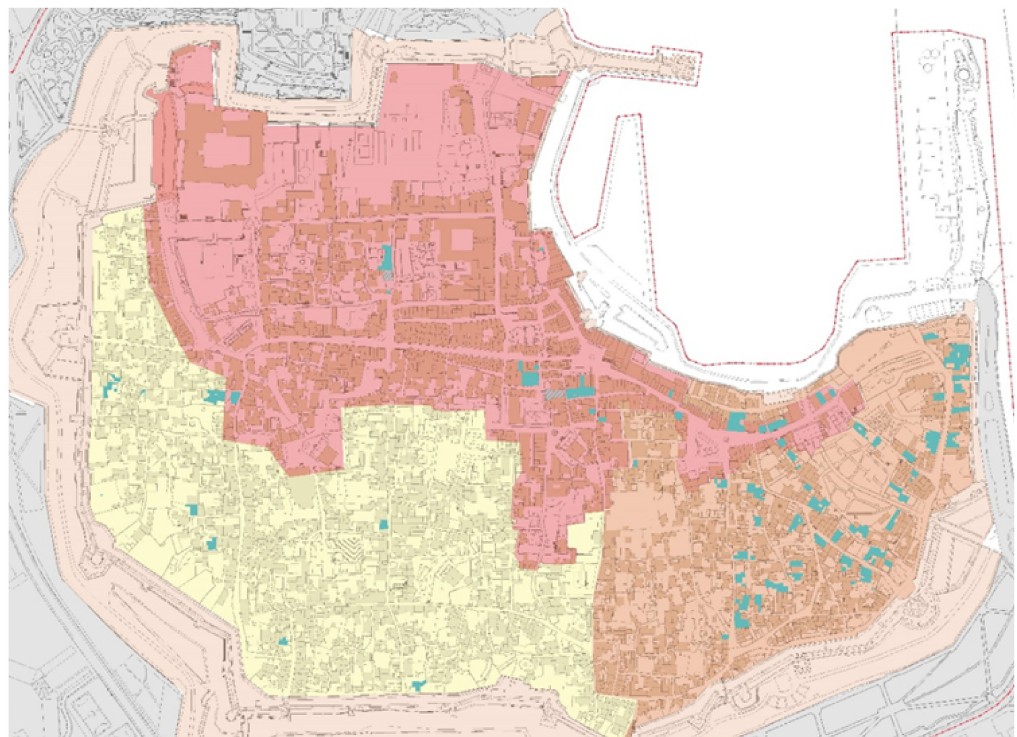

**Figure 14.** Neighborhoods of the Medieval City of Rhodes.

The location criterion mostly affects the housing and the social residency. It is restrictive for the residential uses fitting in the commercial, noisy location of the center of the historic town. The popular, recreation and commercial locations are compatible with commercial use to revitalize arts, the private use of emblematic buildings as well as with tourism services.

(c)    The Existing Use

Initially, for the existing uses (Figure 15), it was decided that they remain the same in order to be compatible with the original use of the constructions. The documentation has to recognize all incompatibilities (Figure 16) preserved today as well as determine the appropriate reuse.

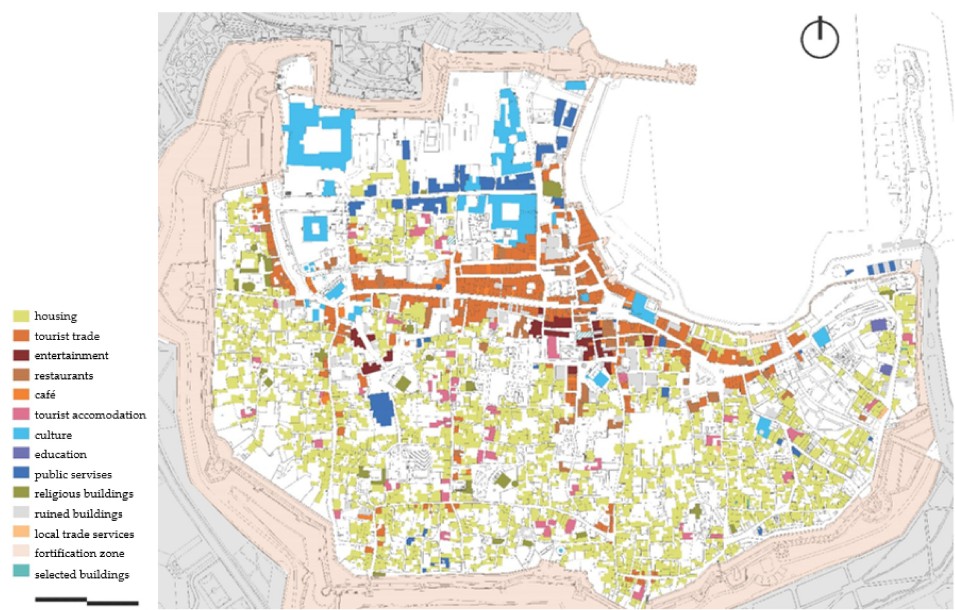

**Figure 15.** Existing land uses of the Medieval City.

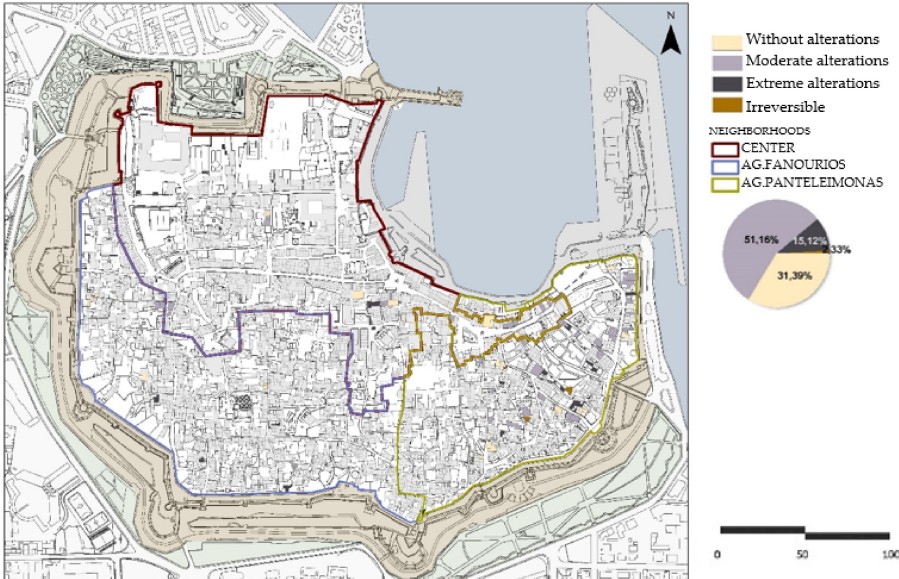

**Figure 16.** State of morphological alterations.

The social housing use is located in the strictly residential zones of the historic city, according to the documentation process. Noisy and touristic uses are rejected from residential zones, in order to maintain regularity to the local community. Another important parameter

is the integration of the social housing in local society of the city. Uses of traditional arts and crafts and also public services are promoted in the existing zone of cultural services, in order to emphasize the extroversion cultural character of the historic city.

(d)   The Coverage

The coverage area of the construction signifies the existence and the scale of the ground floor yard in the plot. The number of floors (first and second floor) reveals the building area of the structure (Figure 17), and thus the exploitable area of the building.

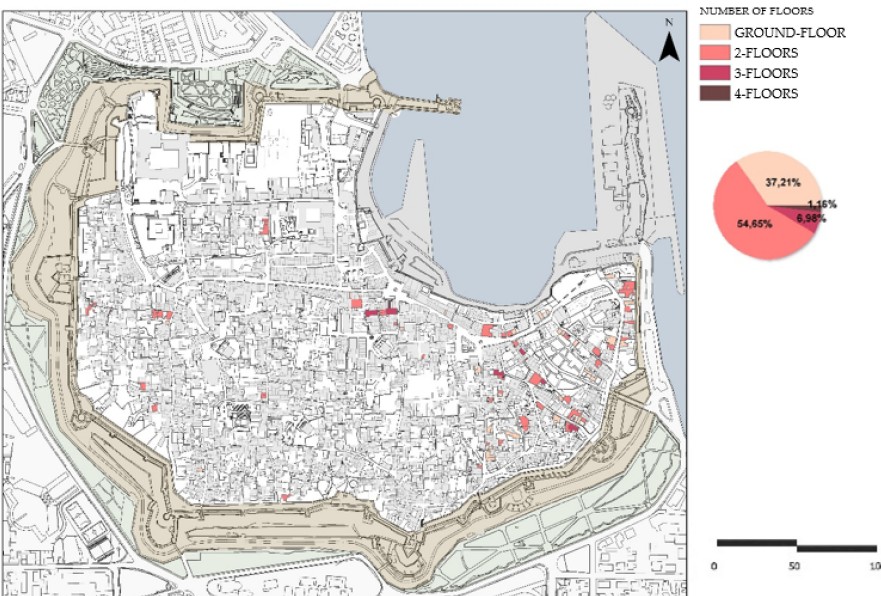

**Figure 17.** Number of floors (ground floor, 1st/2nd/3rd/4th floor).

The coexistence of the criteria of coverage, building area and typology validates the compatibility of reuse proposal. Buildings with small coverage (big yard) are appropriate for workshops of traditional arts and culture but also for recreation services, according to the compatibility of the location.

(e)   The Typology

Typology categories are interlinked with the architectural morphology, the scale and the importance of each building, the coverage and the building area in the plot as well as the roof type.

Vernacular buildings without a remarkable architectural character have been promoted for social residence. The vernacular typology is commonly not located in central areas of the city, according to the documentation. Emblematic buildings of historic significance (Figure 18) are proposed for private uses by international and other organizations to promote extroversion and emphasize the remarkable architectural characteristics. Commonly, these buildings are placed in the center of the Medieval City.

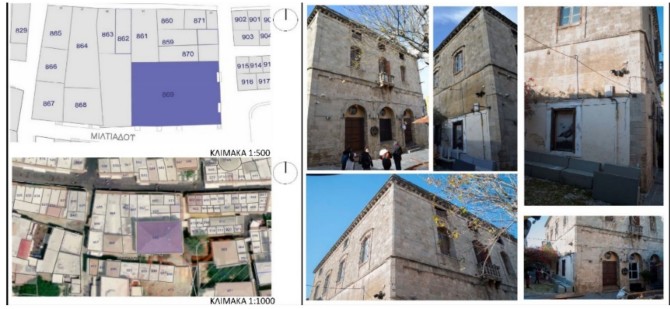

**Figure 18.** Sokratous 4T (parodos), (listed building—evaluation sheet).

(f)    The Conservation Status

The state of conservation categorized as good, medium, poor and collapsed (Figure 9). The conservation status is interlinked with the state of today use (Figure 13). Abandoned buildings are exposed to weather damages, vandalism and natural distractions.

The conservation status in conjunction with morphology assessment affects the criteria of the degree of intervention, especially in the cases of listed monuments (Figure 19). Buildings of a good state of preservation, that do not require special intervention, optimally preserve the existing use.

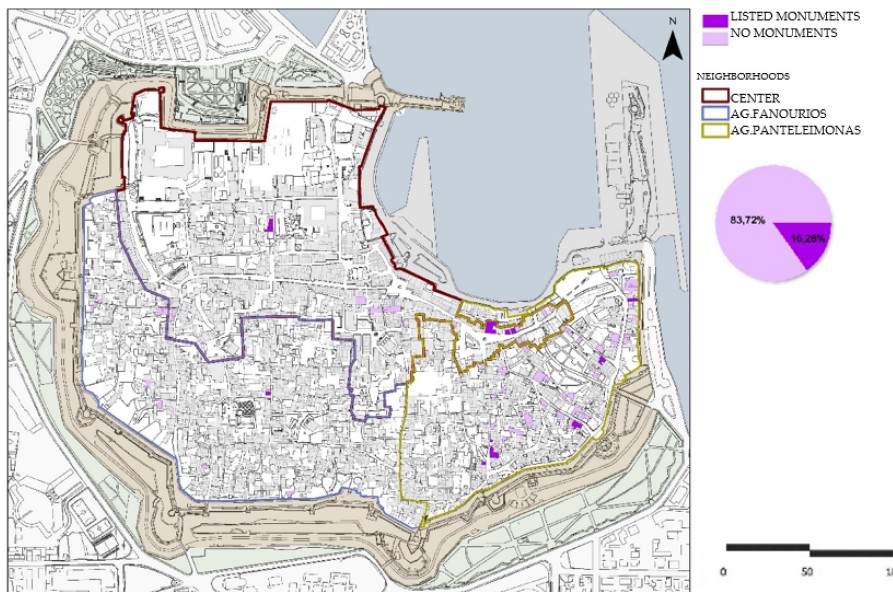

**Figure 19.** Listed monuments of the selected buildings.

Vernacular constructions with a bad conservation status are mostly promoted as social houses (Figure 20).

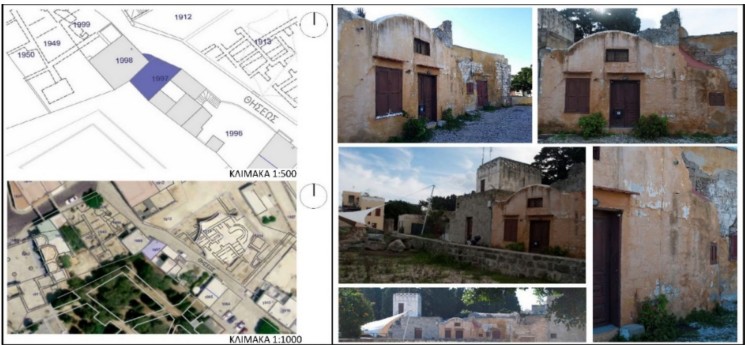

**Figure 20.** Thiseos 08 (listed building—evaluation sheet).

(g)    The Morphology Assessment

Typological categories and morphological assessment are crucial criteria for the compatibility and also for the appropriateness of the reuse proposal. Morphology assessment concerns the façade and the exterior cell of the building, in contrast with the typology concerned with the floor plan. The morphology assessment depends on the individual architectural morphology of every separate building as well as on the historic phase of the construction (Figure 21). Categories with the same morphological character are based on the historic period of construction, which are the medieval, Ottoman, neoclassical, Italian and vernacular (laic).

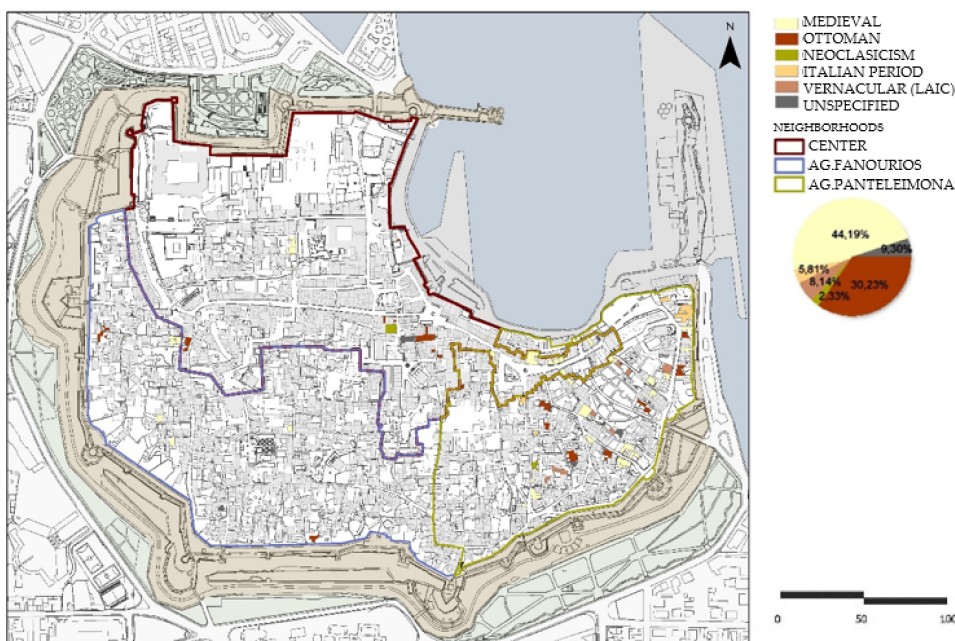

**Figure 21.** Historical period of buildings.

Medieval, Italian and neoclassical morphologies mostly display architectural peculiarities (Figure 22). Vernacular (laic) houses are usually simpler with basic floor plan types and facades without special architectural character.

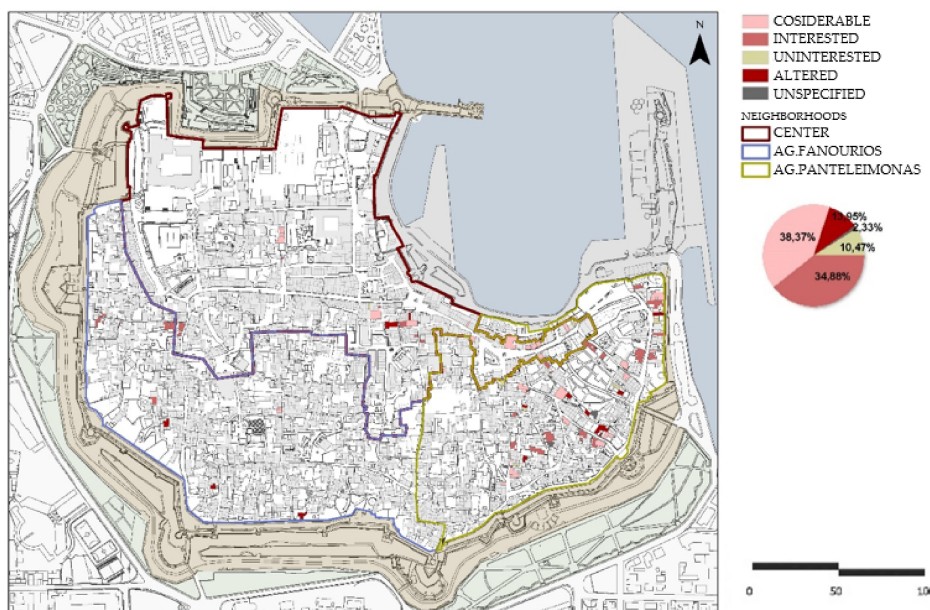

**Figure 22.** Evaluation of selected buildings.

Complex morphologies are observed for neoclassical buildings, which require a high degree of intervention according to the documentation, due to weather damages, the old age of the materials, defacements by incompatible uses and etch. Buildings with no remarkable architectural morphologies are not proposed for recreation or culture services, according to the criteria.

(h)    The Proposed Use

The proposed uses of the project intend to ensure a regular living for the local community and also emphasize the unique character of Medieval city. Therefore, it is crucial to

eliminate degraded areas that are filled with ruined buildings, illegal squats and incompatible land uses. Social housing residents are allocated degraded areas (Figure 23) in order to be compatible with the vernacular character of the current use and to compose a single zone of social housing.

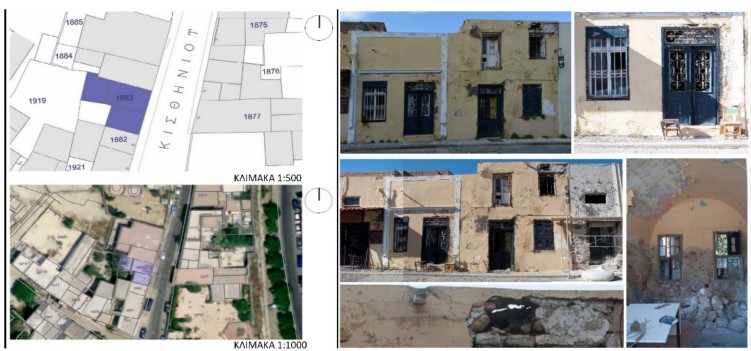

**Figure 23.** Kisthiniou 38–40 (listed building—evaluation sheet).

Cultural uses and workshops of traditional and arts are hosted at emblematic map buildings (Figure 18), according to the documentation. It is not appropriate to place tourism services in buildings with a particular architectural character according to the evaluation criteria. Concerning all commercial uses, the location is crucial for the reuse proposal. Leisure activities, tourism accommodation and recreation services are considered as popular and central uses that give prominence to the extroversion of Medieval City of Rhodes.

(i)      The Proposed Degree of Intervention (Intervention Category)

The degree of intervention is unique for each construction and is totally adapted to the study of the reuse of preservation plan. The proposed degree (Figure 24) is mostly affected by the criteria of typology and morphology assessment. Intervention categories are grouped as follows: (1) absolute conservation; (2) cell restoration; (3) reconstruction; and (4) demolish and rebuild. Buildings that need to undergo major intervention works are easily adaptable to new uses and convenient for radical reconstruction according to the compatibility criteria. On the other hand, constructions with a low degree of intervention, are, mostly proposed for leisure activities and commercial uses should such uses be acceptable according to all the previous criteria. Otherwise, they are promoted as social residence.

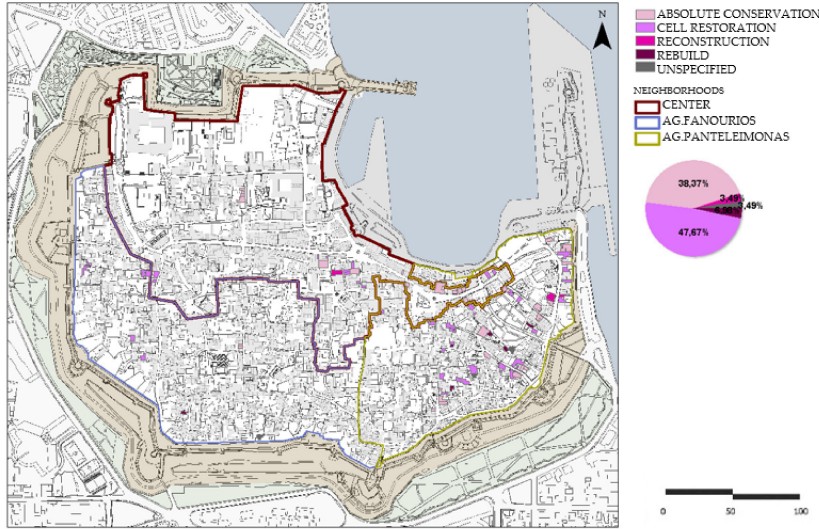

**Figure 24.** Restoration proposal (absolute conservation, cell restoration, reconstruction, demolish and rebuild).

*4.5. Execution and Monitoring the Reuse Project—Threads in the Preservation of the Medieval City of Rhodes (Phase 5)*

The final phase is the execution of the reuse proposal and also the monitoring of the project. Networking between public agencies, local non-profit entities working to preserve the heritage, communities and private investors is needed in order for heritage conservation to be expanded and support broader objectives and longer term goals, assuring that new private investments will be mobilized by public, national and local government incentives. Financing tools are required to support the key areas where cultural heritage losses are escalating as a result of development pressures and hazards [34].

The Medieval City of Rhodes, and more specifically, its fortifications, bastions and ramparts are undergoing intense decay and present pathology phenomena. Aggressive sea salt spray is triggering salt decay and material degradation in synergy with structural pathology phenomena, due to severe earthquakes, under the exercise of lateral thrusts in the contra Scarpa, as well as structural failures due to material losses and the disintegration of the joint lime mortars. The use of incompatible repair materials, such as cement concrete as joint mortars by the De Vecchi restoration during the Italian occupation [67], or the use of incompatible substitution building stones by later interventions, trigger the acceleration of salt decay. The replacement of the decayed pore stones with new stones of smaller porosity enhances the decay phenomena in the surrounding original material at the interface between two stones as visualized by digital image processing and validated by fiber optics microscope (FOM) inspection and scanning electron microscope (SEM) examination. The National Technical University of Athens has continuously monitored and assessed the preservation state of the Medieval fortification and ramparts on an interdisciplinary basis since the early 1980s and is capable of scientifically supporting a sustainable preservation plan.

## 5. Discussion and Challenges

*5.1. Reuse Proposal Implementation*

The adaptive reuse proposal is the result of the implemented methodology that was described in the previous chapter. The innovation of the methodology was the establishment of evaluation criteria as the factors that lead to the adaptive reuse proposal. The proposal as a development program is a theoretical approach that can be established in other historic cities. All the maps and on-site research material are an integral part of research and contribute to scientific knowledge. In Figure 25, the factors that determine the reuse proposal are presented and consist of the key factors for other adaptive reuse proposals.

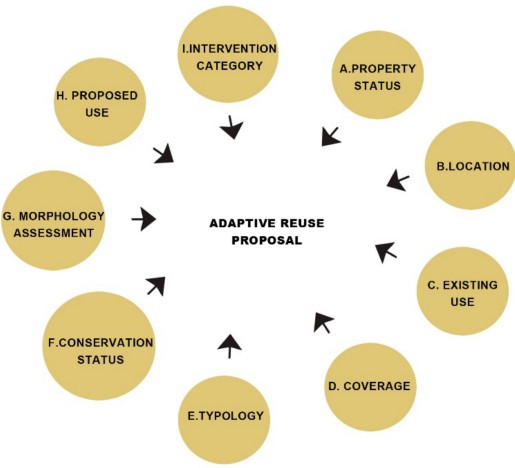

**Figure 25.** Factors of the proposal.

*5.2. Circular Economy*

In order to achieve sustainability, the circular economy could be considered a promising approach [70]. The reuse proposal could contribute to the development of circular

economy initiatives, where resources are used in a more sustainable way. The Medieval City of Rhodes provides the setting for an innovative reuse proposal which develops and promotes, for first time ever, the circular economy in the context of historic cities' sustainable preservation. The presented proposal is based on the renovation and reuse of listed buildings and could contribute to the further development of circular economy initiatives. Its benefits are expected to be prominent in the medieval city fortifications, the moat, the walls, the infrastructure, provision of subsidized public housing and other relevant projects in the City of Rhodes. A total of 300 buildings are comprised within the proposal's scope and thus could be renovated and reused.

Innovation as a key driver towards the circular economy is related to interdisciplinary knowledge-based decision making, "digital" driven preservation, integrated environmental impact assessment and preservation management for resilience enhancement and reconstruction. Hence, the sustainable preservation and future management program of the Medieval City of Rhodes has the potential to implement the concept of a heritage-driven economy by generating revenue whilst contributing to the preservation of the Medieval City, to the social cohesion of the city of Rhodes and to the development of circular economy initiatives [68].

### 5.3. Benefits of the Reuse Proposal

The municipality of Rhodes could implement the described proposal in order to preserve the Medieval City fortifications, building structures in the moat and medieval walls by the sea. In addition, the municipality will fund the provision of social housing to its citizens for a long period.

The reuse proposal could also act as a pioneer for new initiatives and more specifically could assist with (a) studies for special projects and cultural events (Fund of Archaeological Proceeds—TAP); (b) research and training in the context of the framework agreement (Laboratory for the Preservation of Cultural Heritage Monuments—NTUA AIEN); (c) maintenance of the Medieval City's infrastructure (Municipality of Rhodes); (d) upgrade of museums' infrastructure (Fund of Archaeological Proceeds—TAP); (e) structuring, management, strengthening and modernization of projects and activities in the context of the framework agreement for the Medieval City of Rhodes (Cultural Municipal Enterprise and Local Development Enterprise of Rhodes—DERMAE); and finally (f) set up the operation of the International Centre for the Protection, Preservation and Sustainable Development of Heritage Towns.

The Municipality of Rhodes with the reuse proposal could have a comprehensive program for the support of its financially challenged citizens, many of whom reside in the Medieval City of Rhodes. The reuse proposal could additionally supplement the benefits of a future restoration project, by offering many of the renovated buildings available to citizens for a token rent.

This paper underlines the significance of the uniqueness of every historic urban environment. City planning efforts are not replicable because no two cities are alike. Consider the replication problems inherent in a landscape architecture plan: the plan is tailored to a specific site with specific soil, sun and hydrological conditions and these planting prescriptions are hardly relevant to any other site. The city, with its own particular socioeconomic conditions, physical fabrics and history, which significantly change from year to year, faces similar challenges. In other words, city plans are not replicable across time or space—this plan will never be created again, neither in the same jurisdiction or in another [69]. This aspect answers every opposite idea about managing cultural heritage. According to other assessment method about historic cities, such as urban green rating system (UGRS), it is stated that 'it has never been clarified until now how and if they can be used in urban contexts that are bearers of cultural heritage values, containers of elements to be preserved. A lack of a comprehensive review on the role of UGRS for urban planning and renovation in the relation with heritage environments is underlined'. The protocols of UGRS are not specifically designed for historic urban areas [71].

## 6. Conclusions

The respect of interfaces represents a key point in the process of cultural heritage conservation. The improper treatment of historical buildings discontinuities, their loss or alteration, could gradually destroy ancient architectural signs and preclude the possibility to improve the knowledge of ancient buildings in future years [72].

The reuse program, designed taking into account the specificity of the monumental ensemble, aims at the current prevention and future protection of the building and fortification volume, the improvement of the conditions and the resettlement in sustainable terms. The functional reintegration of the selected buildings affects the wider spatial and urban planning and has immediate results in terms of the balancing and harmonious coexistence of permanent and ephemeral (tourist) housing.

The methodology of the development of the studies must be interdependent with a specialized mechanism of control, documentation and recording of the problems, the evaluation of the planning and the required interventions with the securing of financial programs. The reuse strategy presupposes the parallel establishment of mechanisms for monitoring social transformations and the effectiveness of urban redefinitions, with the aim of further analyzing urban phenomena. The interventions aimed at the reuse of buildings, the promotion of the sea fortification, the unification of the beaches, the creation of promenades as well as the arrangement of the traffic and parking are some of the issues that need to be addressed in terms of the drafting and perspective of the general management plan. The promotion and protection of the Medieval City of Rhodes needs a general management plan in order to be treated as a complex case study. The efficiency of the process is the product of a functional intervention planning and the elaboration of the final integrated conservation plan that defines the criteria, goals and parameters of reuse.

Today, thirty years after the enlisting of the Medieval City of Rhodes in the UNESCO World Heritage List, the public interest entities must contribute to the development of a strategic preservation plan for the sustainable preservation and management of the Medieval City of Rhodes and ensure that funds are made available. The historic city of Rhodes acted as a field of interdisciplinary research performed towards all the multilateral aspects of the preservation of the city.

Keeping heritage alive is a perception of sustainable cities, with the active participation of people in the development processes. In such a model of a city development, education becomes a necessity to keep societies informed and research becomes a tool to de-fine new concepts and investigate effective, innovative and compatible planning techniques and materials for the sustainable preservation of historic cities.

**Author Contributions:** D.G. assisted in the design of the methodological approach of the study which consists part of his doctoral research entitled " Methodology of interdisciplinary development and preservation of Cultural Heritage Sites " under the supervision of A.M., conducted and analyzed the evaluation criteria for the reuse proposal and contributed to the writing of the manuscript; Z.K. elaborated the results for her Master Thesis at the NTUA Masters' Inter-Departmental Program for the "Protection of Monuments"—Direction "Materials and Conservation Interventions, entitled "Study of reuse of historic public property buildings in the medieval city of Rhodes" under the supervision of A.M., E.M. (Eleni Maistrou) and S. Augerinou; E.M. (Elli Menegaki) elaborated the results for her Master Thesis at the NTUA Masters' Inter-Departmental Program for the "Protection of Monuments"—Direction "Materials and Conservation Interventions, entitled "Classification, recognition and classification of historic buildings of the Medieval Rhodes city based on multiple criteria and integrated data management geographic information systems" under the supervision of A.M., E.M. (Eleni Maistrou), C.I. and S. Augerinou; E.T. assisted in the analysis of the geospatial information, in the thematic map development and contributed to the writing of the manuscript; A.G. contributed to the finding and gathering of the on-site material. A.M. was responsible for the scientific administration and scientific supervision of the research team, the scientific conceptualization and visualization of the research. All authors have read and agreed to the published version of the manuscript. Credits for the figures that are not adopted by other reference follow the manuscript.

**Funding:** This research received no external funding.

**Institutional Review Board Statement:** Not applicable.

**Informed Consent Statement:** Not applicable.

**Data Availability Statement:** Not applicable.

**Acknowledgments:** The authors would like to thank the President of (H.O.C.RE.D.) Hellenic Organization of Cultural Resources Development (former TAP) and Ministry of Culture Athena Hatzipetrou for the disposal of date for the properties of TAP of Medieval city of Rhodes, while the on-site investigation was taking place and also, the current mayor of Rhodes Antonis Kampourakis, who hosted the NTUA team at the city of Rhodes.

**Conflicts of Interest:** The authors declare no conflict of interest.

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
