# Peer review of "Reuse of Historic Buildings in the Medieval City of Rhodes to Comply with the Needs of Sustainable Urban Development"

_land, doi:10.3390/land11081214_

Round 1

Reviewer 1 Report

Thank you very much for considering my suggestions. The quality of the paper was improved significantly 

Author Response

We would like to thank the reviewers and the editor for their valuable comments and recommendations. We tried to address all the comments and recommendations of the reviewers in a reformed manuscript. The comments of the reviewers are high appreciated and the general structure of the paper has been modified, in order to address all the comments.

Reviewer 2 Report

The paper „Reuse of historic buildings in the Medieval City of Rhodes to comply with the needs of sustainable urban development” illustrates a project for the reuse of selected historic buildings in the Medieval city of Rhodes in Greece. The Authors analyzed how an evaluation of selected buildings leads to an adaptive reuse proposal of land-uses in a historic city, within the certain context of a general sustainable preservation plan. The article is an interesting study in line with current research trends related to the preservation of cultural heritage in the context of sustainable use of space.

The article contains all the elements that should be found in a good scientific article. Particularly to be commended is the very rich editorial page, where the results of the analyses carried out are visualized in several figures. However, I have comments on the content presented in the following chapters and on the originality of the analysis results presented.

1. I ask the authors to clearly indicate in the article what is the author's new contribution to the presented research. When presenting the results, the authors directly point to two Thesises postgr. St - item 10 and 13 in References. Chapter 6, on the other hand, is indicated in its entirety as a reference item 15 from References (the title indicates the same subject matter). The article in its current version looks like a report on the preparation and implementation of a previously created program. The reader can't quite determine what the authors have added to the earlier published studies. In the Results chapter, most of the elements, especially the figures, are directly taken from other studies.

2. The Introduction chapter needs to be expanded. In the current version, it is difficult to find in it a demonstration of the scientific background of the research conducted.

3. Chapter 2 Materials and methods contains an indication of the purpose of the research and a brief description of the process presented in the article. Subsections 2.1-2.3 are more fitting as a missing presentation of the current state of knowledge in the topic under study. There is a lack of research methodology, an indication of the research methods used with their justification.

4. The results are presented clearly, richly illustrated. I only have the previously indicated doubts about their originality in that article.

5. Chapter 6 Discussion and challenges presents the results and not their discussion. Please refer, for example, to other studies of this type conducted by other Authors elsewhere and indicate in what elements the proposed solution is better, what is innovative about it. In addition, that earlier mentioned reference next to the chapter title is questionable.

6. Fig. 9-10 - what are the referrals: B2, B3 and B4?

The article needs to be refined and rewritten. Please pay particular attention to Comments 1. and 3. - these two elements are the most questionable.

Round 2

Reviewer 2 Report

In response to my comments in the first review, the Authors made significant corrections and additions to the article. In the current version, it is clearly indicated what is their author' s contribution to the article and what constitutes new proposals. The authors' clarification of my doubts about the use of earlier proposals is completely satisfactory to me.

I wish you good luck in your further scientific work.

This manuscript is a resubmission of an earlier submission. The following is a list of the peer review reports and author responses from that submission.

Round 1

Reviewer 1 Report

The paper illustrates the project for the reuse of the Medieval City of Rhodes. It can be considered a case study paper, not a research paper as it is a description of the case study without an innovative approach. The paper is really focused on the case study description without any reference to sustainable development and reuse of urban heritage. Probably a complete scenario in this topic can help to enlarge the sims and the vision of the paper. The recent publication https://doi.org/10.1016/j.rser.2022.112324Can suggest you new direction in the research in urban heritage, related to techniques, procedures, green rating systems, regenerative design. Nahoum C., Urban Planning Conservation and Preservation, McGraw-Hill: New York, 2001. Sustainable Historic Towns: Urban Heritage – Good for the Climate!, PROJECT REPORT 2011-2012, Dag Arne Reinar and Frederica Miller (Editors), 2012. You can find other publications on urban heritage preservation. Some parts are  not referenced for example the Factors that affect the revitalization strategy of a Mediterranean historic city. There is a huge literature on it, and some papers have opposite ideas. Reference better the osier to have a more scientific explanation. The methodological approach is not described. For example you identified some problems. It is not clear the approach you used for their  identification. The same for identifying the principles for reuse. Do you use specific focus groups? Do they derive from urban regulations?

Reviewer 2 Report

First of all, I must admit that the topic of the paper does not seem very suitable for the purposes of the LAND journal. In fact, the paper deals with the reuse of the medieval buildings of Rhodes in the urban environment and the approach is essentially that of Heritage studies. Therefore (I do not know if it is possible) I would propose to publishers and authors to think about moving the article for the MDPI journal HERITAGE, in my opinion much more appropriate to the topic of the article.

The article is well written and explains the objectives and ideas of the article well.

Line 15: Why you use Medieval City and not Medieval city?

Line 33: please to provide some reference for people who are not familiar with Rhodes history.

Line 36: why are you using settlement here?

Line 43: in the paper you use Medieval and sometimes medieval, please to choose a coherent form.

Line 49: please to add page number of this sentence

Line 51: “Cities of the past”. Not sure about the meaning of this sentence. Please to provide another formula.

  • FIGURES FROM 1 TO 15 have not a reference inside the text, it must be corrected

Line 76: “according to the restorative disciplinary”: provide reference

Line 109: in theory this figure must be number 2, please to check and correct the numeral sequence of the figures

Line 109 and 111: why are you using here capital letters?

Line 114-115: “historic sites integration into city context with a web of pedestrian zones”. I don’t understand this sentence, “web”?

Line 122-128: for a scientific paper you must insert all the reference about geology and history of Rhodes.

Line 133: what do you mean for “chivalric period”?

Line 137, 140, 154: why are you using here capital letters?

Line 156-167: also here you need to add references

Line 218: please to provide reference “De Vecchi restoration during the Italian occupation”

Line 293: please to provide reference “According to Ms. Moropoulou”

Line 236: The location of the selected building is not clear in the map of figure 10. So indicate it with numbers or something else.

Line 321: The legend of figure 12 is cut and the map needs a bar scale and a northern arrow

Line 331: In the list of Evaluation criteria after each title you use a “,” but sometimes not, so please to homogenize everything.

Line 454: it looks there is a problem in paragraph spacing

Line 502: Figure 15 is irrelevant, remove it

Line 521: The legend of figure 15 (again here there is an error in number sequence of the figures) needs a bar scale and a northern arrow

Line 631-656: Figure B1-B12 needs a bar scale

Line 763-799: please to format the reference according to the journal guidelines

Round 2

Reviewer 1 Report

The novelty of the paper was not improved. The paper is a case study discussion not an original paper on a specific methodology. The added data in the section 2.1 are not integrated in the paper. It seems a general discussion on similar cities without understanding why you select these cities, the similarities, and without adding new comparisons on it. Please consider again the previous comments. I mean that you should discuss urban conservation aspects considering other theories, urban green rating systems, and other assessment methods. 
